# The Synergy Dilemma of Long-CoT SFT and RL: Investigating Post-Training Techniques for Reasoning VLMs

**Jierun Chen**[*1], **Tiezheng Yu**[* 1], **Haoli Bai**[#1] **Lewei Yao**[1], **Jiannan Wu**[1], **Kaican Li**[2], **Fei Mi**[1],
**Chaofan Tao**[1], **Lei Zhu**[1], **Manyi Zhang**[1], **Xiaohui Li**[1], **Lu Hou**[1], **Lifeng Shang**[1], **Qun Liu**[1]
[1]*Huawei Technologies*, [2]*HKUST*

**Reviewed on OpenReview:** `https://openreview.net/forum?id=XPML8UGIO4`

## Abstract

Large vision-language models (VLMs) increasingly adopt post-training techniques such as long chain-of-thought (CoT) supervised fine-tuning (SFT) and reinforcement learning (RL) to elicit sophisticated reasoning. While these methods exhibit synergy in language-only models, their joint effectiveness in VLMs remains uncertain. We present a systematic investigation into the distinct roles and interplay of long-CoT SFT and RL across multiple multimodal reasoning benchmarks. We find that SFT improves performance on difficult questions by in-depth, structured reasoning, but introduces verbosity and degrades performance on simpler ones. In contrast, RL promotes generalization and brevity, yielding consistent improvements across all difficulty levels, though the improvements on the hardest questions are less prominent compared to SFT. Surprisingly, combining them through two-staged, interleaved, or progressive training strategies, as well as data mixing and model merging, all fails to produce additive benefits, instead leading to trade-offs in accuracy, reasoning style, and response length. This "synergy dilemma" highlights the need for more seamless and adaptive approaches to unlock the full potential of combined post-training techniques for reasoning VLMs. Code, dataset, and fine-tuned models are available at https://github.com/JierunChen/SFT-RL-SynergyDilemma.

## 1 Introduction

Large language models (LLMs) like OpenAIs o1/o3 (Jaech et al., 2024) and DeepSeek-R1 (Guo et al., 2025) have demonstrated remarkable reasoning abilities by *thinking before answering*. These models go beyond mere pattern matching, exhibiting sophisticated cognitive behaviors like multi-step planning, reflection, error correction, as well as summarization (Gandhi et al., 2025; Wang et al., 2025c). This reasoning capability is primarily enabled by two core post-training techniques: Supervised Fine-Tuning (SFT) on long chain-of-thought (CoT) data (Muennighoff et al., 2025; Moshkov et al., 2025; Sun et al., 2024), and Reinforcement Learning (RL) with verifiable feedback (Lyu et al., 2025; Luo et al., 2025). In language-only domains, these methods often show synergistic effects, yielding substantial improvements on complex reasoning benchmarks when applied sequentially or iteratively (Liu et al., 2025; Team et al., 2025; Yeo et al., 2025).

This paradigm has naturally motivated researchers to apply similar paradigm to large vision-language models (VLMs) in pursuit of comparable gains in multimodal reasoning (Huang et al., 2025; Zhang et al., 2025; Wang et al., 2025a). However, the results have been inconsistent and controversial. On one hand, some findings suggest that even small-scale SFT on long-CoT traces can elicit step-by-step multimodal reasoning and improve accuracy on multimodal math benchmarks (Du et al., 2025). On the other hand, some studies report that SFT, even when followed by RL, can degrade performance (Chen et al., 2025a). These discrepancies point to a complex and still poorly understood interplay between training strategies in the multimodal domain.

---

*Equal contribution; # Corresponding author: `baihaoli@huawei.com`.

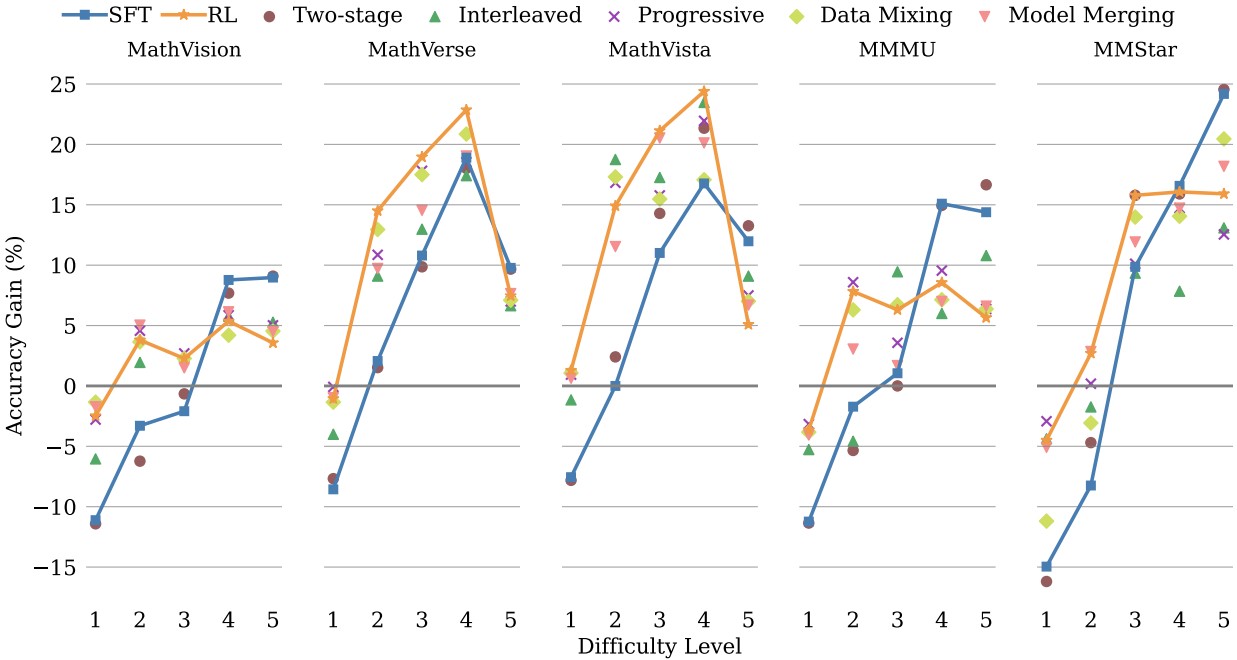

Figure 1: Accuracy gains from various post-training techniques across five difficulty levels (L1, easy to L5, hard) on five multimodal reasoning benchmarks. Long-CoT SFT boosts Qwen2.5-VL-7B on harder questions but hurts easier ones, while RL yields steady gains across the board. Hybrid strategies consistently trade off strengths rather than achieving true synergy.

In this work, we conduct a systematic study, covering diverse post-training paradigms, comprehensive evaluations, and varied models, to answer two key questions:

1) *What unique roles do long-CoT SFT and RL play in shaping the reasoning abilities of VLMs?*

2) *Can we effectively combine them to realize the best of both worldsstructured reasoning and robust performance?*

To answer the first question, we zoom in current multimodal reasoning benchmarks through the lens of question difficulty, a factor that has often been overlooked. Unlike textual reasoning benchmarks such as AIME25, MATH500 (Hendrycks et al., 2021), and GPQA (Rein et al., 2024), which emphasize logically demanding and difficult tasks, current multimodal reasoning benchmarks like MathVista (Lu et al., 2023), MathVerse (Zhang et al., 2024a), and MMMU (Yue et al., 2024) contain a large proportion of simple questions focused on perception and fine-grained visual understanding rather than complex cognitive reasoning. After categorizing benchmark questions by difficulty, we find that **long-CoT SFT improves performance primarily on hard questions** but degrades it on easier ones by introducing unnecessary verbosity and overthinking. In contrast, **RL offers steady gains** across questions through concise responses and better generalization. That said, its gains on the hardest questions are less significant than those achieved by SFT.

Building on these insights, we then explore several strategies for combining SFT and RL, including two-stage, interleaved, and progressive training, as well as data mixing and model merging. These strategies differ in timing, adaptability, and method of combination. Two-stage and interleaved training focus on when each method is used: the former separates SFT and RL into distinct phases (first SFT, then RL), while the latter interleaves them step by step. Progressive training adds adaptability, fading hints in SFT over time to smoothly transition toward pure RL. Meanwhile, data mixing blends data distilled from SFT and RL for a new round of fine-tuning, while model merging directly combines fine-tuned models parameters via interpolation. Despite their promise, they all hit a **synergy dilemma: efforts to fuse long-CoT SFT and RL often produce trade-offs rather than true complementarity**, as shown in Fig. 1.

For instance, interleaved training balances performance but cannot surpass standalone RL, and data mixing preserves neither SFTs strength on the MathVision benchmark nor RLs broad gains across other benchmarks.

By surfacing these insights, we provide new clarity on various post-training techniques for reasoning VLMs. We demonstrate that synergy between SFT and RL is fragile. Achieving it requires not just method stacking, but nuanced control over adaptivity, compatibility, and difficulty-awareness.

## 2 Distinct Effects of Long-CoT SFT and RL

In this section, we investigate the distinct effects of long-CoT SFT and RL in enhancing multimodal reasoning for VLMs. We begin by analyzing how long-CoT SFT influences performance depending on the modality, scale, and reasoning quality of the training data, as discussed in Sec. 2.1. We then examine RL training dynamics, highlighting the importance of KL regularization and the necessity of incorporating simple questions (see Sec. 2.2). Finally, we present in Sec. 2.3 a systematical comparison between SFT and RL, revealing that SFT tends to benefit harder questions through verbose, structured reasoning, while RL yields steadier improvements with concise responses. Together, these findings offer a nuanced view of how each method contributes to reasoning capabilities in multimodal settings.

### 2.1 Long-CoT SFT

Supervised fine-tuning (SFT) with long chain-of-thought (CoT) data has proven effective for language models (Guo et al., 2025), particularly in the mathematical field (Muennighoff et al., 2025). However, its efficacy for multimodal reasoning remains debated. For example, fine-tuning Qwen2.5-VL-7B with an R1-Onevision reasoning dataset yields marginal improvements on MathVision (Wang et al., 2024) and even declines on MathVerse (Zhang et al., 2024a), regardless of increased model size or data scale (Chen et al., 2025b). This raises the question of whether long-CoT SFT offers any tangible benefits for multimodal reasoning. To answer this, we access the effectiveness by varing the data sources and modalities.

**Data.** VLM reasoning is typically regarded as textual reasoning conditioned on visual inputs. Therefore, we hypothesize that SFT with purely textual long-CoT data can enable multimodal reasoning. We use the s1 dataset with 1k diverse and challenging questions (Muennighoff et al., 2025). It has two versions, s1 and s1.1-R1, distilled from Gemini2.0-flash (Google, 2024) and DeepSeek-R1 (Guo et al., 2025), respectively. To study the effect of data modality, we construct a multimodal reasoning dataset by distilling from our s1.1-R1 SFT-ed model on MM-Eureka queries (Meng et al., 2025). We generate eight responses per query, retain the shortest correct one, and obtain our Eureka-Distill with 34k samples. For each query, we append a reasoning instruction: `Please reason step by step within <think> </think> tags, and put your final answer within \boxed{}.`

**Training settings.** We employ LLaMa-Factory (Zheng et al., 2024) as the training framework, and Qwen2.5-VL-Instruct (Bai et al., 2025) as the main baseline model for its strong performance and broad adoption in recent work (Wang et al., 2025a; Yang et al., 2025a; Huang et al., 2025). The ViT visual encoder and MLP connector remain frozen during training, as this way compares favorably to unfreezing them. We use a learning rate of $1 \times 10^{-5}$ and a batch size of 32. We train on s1 and s1.1 for 15 epochs and on Eureka-Distill for 5 epochs. Checkpoints are saved after each epoch.

**Evaluation settings.** We evaluate models using VLMEvalKit (Duan et al., 2024) on both mathematical and multi-disciplinary reasoning benchmarks, including MathVision test (Wang et al., 2024), MathVerse test mini (Zhang et al., 2024a), MathVista test mini (Lu et al., 2023), MMMU val (Yue et al., 2024), and MMStar val (Chen et al., 2024). We append the aforementioned reasoning instruction to the benchmark questions when evaluating fine-tuned models. For answer extraction and judgement, we use rule matching and Qwen2.5-VL-32B as the judge. For inference, we enable sampling using a temperature of 0.6, a top-p of 0.95, a top-k of 20, and a maximum generation length of 24k. Results are averaged over 4 runs to mitigate statistical variance. To ensure fair comparison under one framework, we reproduce Qwen2.5-VL-7B results. For fine-tuned models, we report their best checkpoint results. We maintain consistent evaluation settings throughout the paper.

**Results and analysis** are summarized as follows:

- **Using just 1k textual data, s1.1-R1 improves multimodal reasoning** across four benchmarks: MathVision, MathVerse, MathVista, and MMMU val, with only a slight trade-off on MMStar compared to the baseline. This is particularly notable given that s1.1-R1 consists purely of textual long-CoT traces, yet outperforms Eureka-Distill, a much larger multimodal dataset with 34k samples (see Tab. 1). These results suggest that high-quality reasoning traces, even without visual input, can effectively transfer to multimodal tasks due to the underlying alignment between language and vision in VLMs.

- **Long-CoT SFT improves more prominently on difficult benchmarks** that inherently require deeper reasoning, such as MathVision and MathVerse. As illustrated in Fig. 3, performance gains correlate positively with response length, indicating that longer reasoning chains are more beneficial for complex queries. This pattern highlights the utility of long-CoT supervision in teaching models to break down and solve harder problems step-by-step.

- **Long-CoT does not guarantee performance gains.** The effectiveness of SFT depends heavily on the quality of the reasoning traces. For example, when fine-tuning with s1-Gemini2, despite using the same query set as s1.1-R1 and generating responses that are 815 times longer than the baseline (see Fig. 2), model performance deteriorates across nearly all benchmarks (see Tab. 1). These findings underscore that verbosity alone is insufficient; the quality of the reasoning steps is crucial for realizing effective test-time scaling.

Table 1: Comparison of SFT with different textual and visual long CoT data. M. is short for Math.

| Model | Modality | Data | M.Vision | M.Verse | M.Vista | MMMUval | MMStar | Avg. |
|---|---|---|---|---|---|---|---|---|
| Qwen2.5-VL-7B | – | – | 26.1 | 42.3 | 66.4 | 53.5 | 63.2 | 50.3 |
| + SFT w/ s1-Gemini2 | Text | 1k | 25.2 | 41.8 | 66.8 | 50.0 | 59.5 | 48.7 |
| + SFT w/ s1.1-R1 | Text | 1k | 30.6 | 48.1 | 67.6 | 54.4 | 61.8 | 52.5 |
| + SFT w/ Eureka-Distill | Visual | 34k | 29.7 | 47.2 | 65.6 | 53.6 | 60.8 | 51.4 |

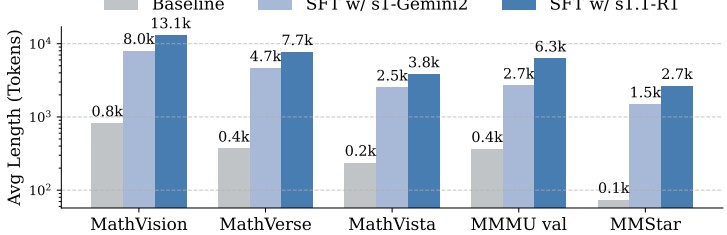

Figure 2: Response length comparison.

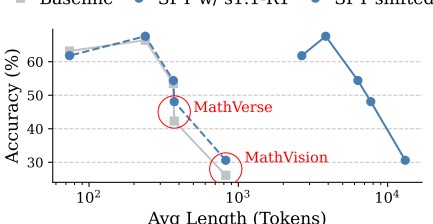

Figure 3: Acc. by response length.

## 2.2 RL

Unlike the above SFT relying on external reasoning traces, reinforcement learning (RL) focuses on enabling models to self-explore and self-improve by interacting with the environment and receiving feedback signals. Notably, the Group Relative Policy Optimization (GRPO) algorithm (Shao et al., 2024) simplifies the training pipeline by eliminating the need for a value model, making the process more efficient and scalable. Mathematically, let $Q$ be the query set and $\{o_1, o_2, \cdots, o_G\}$ be the sampled outputs from the old policy model $\pi_{\text{old}}$. GRPO optimizes the following objective:

$$\mathcal{J}_{\text{GRPO}}(\theta) = \mathbb{E}_{q \sim Q, \{o_i\}_{i=1}^G \sim \pi_{\theta_{\text{old}}}} \left[ \frac{1}{G} \sum_{i=1}^{G} \frac{1}{|o_i|} \sum_{t=1}^{|o_i|} \min \left( r_t(\theta) \hat{A}_{i,t}, \, \text{clip}(r_t(\theta), 1-\epsilon, 1+\epsilon) \hat{A}_{i,t} \right) - \beta D_{\text{KL}}(\pi_\theta \| \pi_{\text{ref}}) \right].$$

Here, $A_i = \frac{r_i - \text{mean}(\{r_1, r_2, ..., r_G\})}{\text{std}(\{r_1, r_2, ..., r_G\})}$ is the estimated advantage using a group of rewards $\{r_1, r_2, \ldots, r_G\}$, $\beta$ is the KullbackLeibler (KL) coefficient contorling the deviation of current model $\pi_\theta$ from the reference model $\pi_{\text{ref}}$, $r_t(\theta)$ is for importance sampling, and $\epsilon$ is the clipping hyper-parameter.

Building on this algorithm, some studies (Yu et al., 2025; Chen et al., 2025c) have proposed removing the KL term for unconstrained exploration. Additionally, it has become common practice to pre-filter or dynamically filter easy samples where the model consistently predicts all answers correctly (Peng et al., 2025; Yu et al., 2025; Wang et al., 2025b). We examine these two variations as they can play a crucial role in model training and performance.

**Training Settings.** Using the Verl framework (Sheng et al., 2024), we train models on multimodal queries from Eureka-Distill as described in Sec. 2.1, with the following settings: a learning rate of $1 \times 10^{-6}$, batch size of 128, mini-batch size of 64, temperature of 1, rollout number of 8, maximum completion length of 4k tokens, and 2 training epochs. For reward functions, we combine two types of rewards: accuracy and format. The accuracy reward is set to 0.9 for correct answers and 0 otherwise, while the format reward is set to 0.1 for correctly formatted responses and 0 otherwise. We use the same reasoning instruction as previously used for SFT fine-tuning: `Please reason step by step within <think> </think> tags, and put your final answer within \boxed{}.`

**Results and analysis** are summarized as follows:

- **The KL term stabilizes the training process.** Without it, the training reward collapses after step 300, and the model exhibits lower entropy and dramatic fluctuations in response length (see Fig. 4). Tab. 2 also shows that KL regularization improves accuracy by a clear margin across all benchmarks.

- **Simple questions matter for maintaining baseline performance.** Without them, the model's ability to handle simple questions could deteriorate after RL fine-tuning. To validate this, we remove the easiest questions on which the baseline model predicts correctly across all 8 rollouts. Results in Tab. 3 show that RL with these easiest questions leads to higher accuracies. This is because, although the advantages for these easiest questions become zero after GRPO normalization, they still influence the training process via the KL loss, ensuring consistently high accuracy on these questions (see Fig. 5).

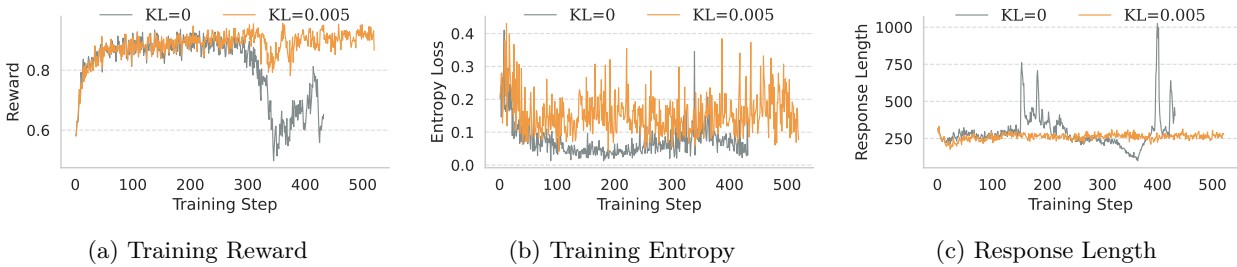

(a) Training Reward           (b) Training Entropy          (c) Response Length

Figure 4: Training dynamics comparisons. Without KL regularization, RL training suffers from reward collapse, lower entropy, and more dramatic fluctuations in response length.

Table 2: Abalation of the KL term for RL training. M. is short for Math.

| Model | M.Vision | M.Verse | M.Vista | MMMUval | MMStar | Avg. |
|---|---|---|---|---|---|---|
| Qwen2.5-VL-7B | 26.1 | 42.3 | 66.4 | 53.5 | 63.2 | 50.3 |
| + RL, KL=0 | 27.7 | 47.0 | 71.5 | 54.8 | 63.5 | 52.9 |
| + RL, KL=0.005 | 29.0 | 52.1 | 72.6 | 55.1 | 66.5 | 55.1 |

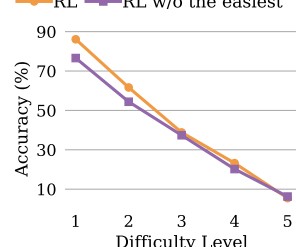

Table 3: Ablation of retaining the easiest questions during RL.

| Model | M.Vision | M.Verse | M.Vista | MMMUval | MMStar | Avg. |
|---|---|---|---|---|---|---|
| Qwen2.5-VL-7B | 26.1 | 42.3 | 66.4 | 53.5 | 63.2 | 50.3 |
| + RL w/o easiest | 26.6 | 47.1 | 71.4 | 54.0 | 64.8 | 52.8 |
| + RL | 29.0 | 52.1 | 72.6 | 55.1 | 66.5 | 55.1 |

Figure 5: Accuracy gap differs across questions of varying difficulty levels on MathVision.

## 2.3 Comparisons of Long-CoT SFT and RL

With optimized data sources and training configurations, both long-CoT SFT and RL yield partial or full improvements in model accuracy. However, the extent of these improvements varies across different benchmarks, as evidenced in Tab. 4. This disparity motivates a granular comparison between SFT and RL fine-tuned models. For fair comparisons, we use the same Eureka-Distilled dataset.

**SFT excels at the most difficult questions, while RL provides steadier, broader improvement.** To analyze performance by difficulty, we categorize all benchmark questions into five levels (easy to hard), based on the baseline models pass rate $P$ across 16 independant runs: level 1 ($P \geq \frac{12}{16}$), level 2 ($\frac{8}{16} \leq P < \frac{12}{16}$), level 3 ($\frac{5}{16} \leq P < \frac{8}{16}$), level 4 ($\frac{2}{16} \leq P < \frac{5}{16}$), and level 5 ($P < \frac{2}{16}$). We also provide a more granular breakdown in Appendix Sec A.1, using 17 pass rate bins (from $\frac{0}{16}$ to $\frac{16}{16}$). From Fig. 7, we see that the accuracy gain from SFT starts negative at level 1, climbs to positive gains at higher levels, and ultimately surpasses RL at the most difficult levels, 4 and/or 5. In contrast, RL provides more consistent improvements across all difficulty levels and benchmarks.

**SFT injects rich yet verbose reasoning tokens, while RL preserves concise responses but under-utilizes structured resoning.** To better understand how these fine-tuning methods shape model responses, we measure token-level KL divergence before and after fine-tuning. As illustrated in Fig. 6, the SFT-ed model consistently exhibits higher KL divergence at sentence beginnings. Those darker tokens highlight what we call reasoning pivotal tokens. These include structural words like first, second, and next; logical connectors such as then, alternative, and because; action words like let, consider, and verify; and meta-thinking phrases like wait, hmm, and maybe. After completing the thinking process, the SFT-ed model first provides a summary before giving the final answer. However, the average response length is more than $10\times$ longer than that of the baseline and RL-tuned models (see Fig. 8), incurring the overthinking risk. In contrast, RL-tuned models remain close to the baseline, with little change in token distribution (see Fig. 6) and a continued preference for brief, to-the-point answers (see Fig. 8) with only occasional use of reasoning words (see Tab. 5).

Table 4: Comparison of SFT and RL with Eureka-Distill as the training set.

| Model | MathVision | MathVerse | MathVista | MMMUval | MMStar | Avg. |
|---|---|---|---|---|---|---|
| Qwen2.5-VL-7B | 26.1 | 42.3 | 66.4 | 53.5 | 63.2 | 50.3 |
| + SFT | 29.7 | 47.2 | 65.6 | 53.6 | 60.8 | 51.4 |
| + RL | 29.0 | 52.1 | 72.6 | 55.1 | 66.5 | 55.1 |

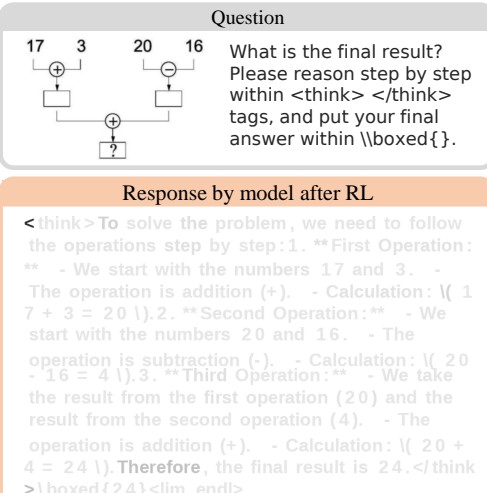

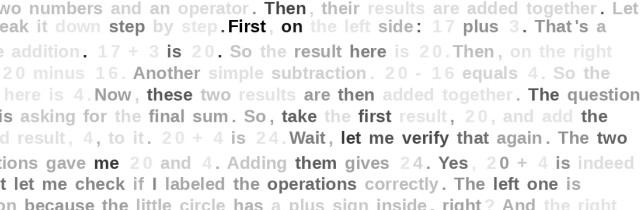

Figure 6: Illustration of token-level KL divergence, where darker tokens indicate larger divergence.

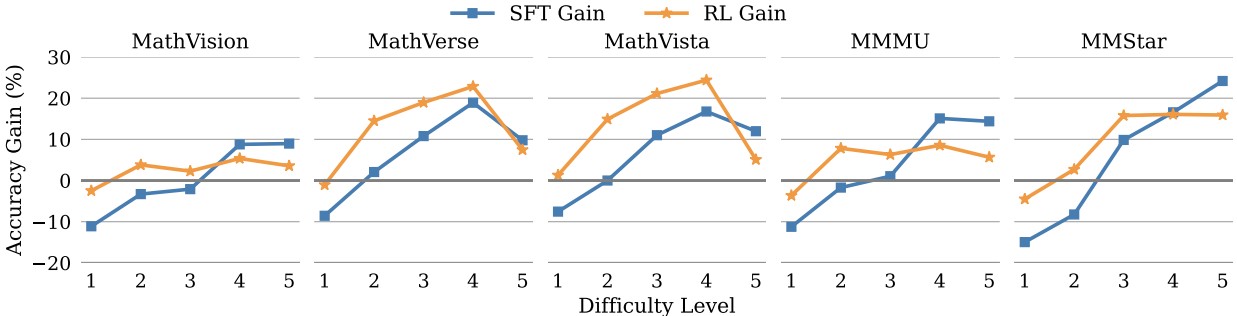

Figure 7: Accuracy gain of SFT and RL across 5 difficulty levels on 5 benchmarks.

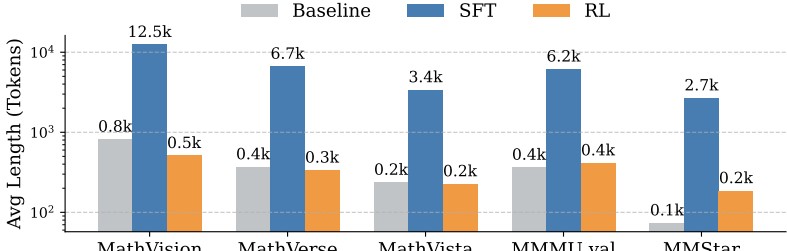

Figure 8: Response length comparison.

Table 5: Frequency of reasoning words evaluated on MathVision.

| Word | Baseline | SFT | RL |
|---|---|---|---|
| "wait" | 0 | 249091 | 0 |
| "check" | 790 | 18501 | 671 |
| "mistake" | 151 | 4345 | 7 |
| "alternative" | 0 | 96815 | 1 |
| "however" | 917 | 36625 | 764 |

## 3 The Synergy Dilemma of Long-CoT SFT and RL

Long-CoT SFT and RL each bring unique strengths and weaknesses in response styles, efficacy, and efficiency in solving problems of varying difficulty. The key question is: ***how can we integrate their strengths to create synergy?*** As shown in Fig. 9, we approach this by exploring several dimensions, including training alternation, data mixing, and model merging. For training alternation, we examine strategies such as the popular two-stage SFT and RL, interleaved SFT and RL, and progressive SFT and RL. Through systematic exploration, we uncover a fundamental ***synergy dilemma for reasoning VLMsLong-CoT SFT and RL often behave more like a trade-off than a perfect complement.*** In the following sections, we dive deeper into each method and evaluate their ineffectiveness in bridging this gap.

### 3.1 Training Alternation

**Two-stage SFT & RL.** Starting from the best checkpoint obtained after SFT on the Eureka-Distill dataset for 4 epochs, we apply RL fine-tuning using the GRPO algorithm. While the experimental setup matched the optimal configuration in Sec. 2.2, we extend the maximum new generation length N from 4k to 16k to accommodate the models longer outputs. This adjustment is necessary because, with N=4k, the clip ratio of model outputs surges to 20% early in training, causing instability and crashes as visualized in Fig. 11. Results in Tab. 6 show that the two-stage SFT and RL approach fails to improve performance over SFT alone, with the average accuracy across five benchmarks stagnating at 51.4%. The model appears to have overfit to the paradigm established during SFT fine-tuning. We have also attempted reducing the number of preliminary SFT epochs to 1, but this does not alleviate the issue, highlighting the difficulty of overcoming SFT-induced overfitting or catastrophic forgetting though a subsequent RL fine-tuning.

**Interleaved SFT & RL.** To mitigate the risk of overfitting and uneven task performance after SFT, we employ an interleaved SFT and RL strategy designed to balance imitation and exploration. Importantly, we do not apply interleaved training to all samples. Applying SFT loss indiscriminately would cause it to dominate the optimization process, thereby slowing or even reducing training rewards and validation accuracy, and resulting in a substantial increase in response lengths (see Fig. 12). Instead, we apply the SFT loss exclusively to questions with zero pass rates, while the RL loss is applied to the rest. This approach aligns

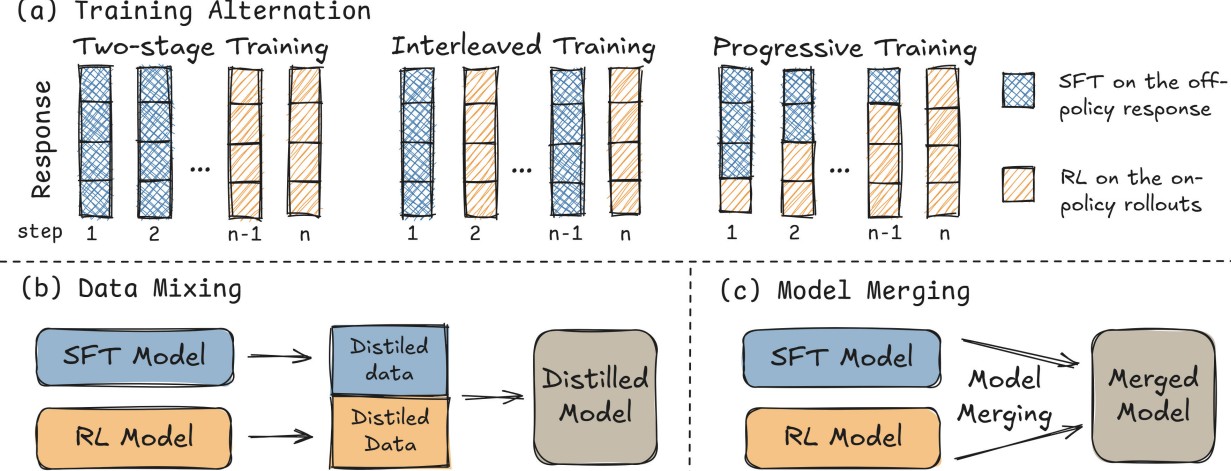

Figure 9: Attempts to integrate SFT and RL, including training alternation, data mixing, and model merging.

Table 6: Acuracy comparison of various attempts for SFT-RL synergy. Values are reported as mean accuracy ± 95% confidence intervals.

| Model | MathVision | MathVerse | MathVista | MMMUval | MMStar | Avg. |
|---|---|---|---|---|---|---|
| Qwen2.5-VL-7B | 26.1 ± 0.40 | 42.3 ± 0.18 | 66.4 ± 0.91 | 53.5 ± 0.51 | 63.2 ± 0.40 | 50.3 ± 0.28 |
| + SFT | 29.7 ± 0.72 | 47.2 ± 0.19 | 65.6 ± 1.42 | 53.6 ± 0.76 | 60.8 ± 0.94 | 51.4 ± 0.46 |
| + RL | 29.0 ± 0.55 | 52.1 ± 0.31 | 72.6 ± 0.44 | 55.1 ± 0.58 | 66.5 ± 0.72 | 55.1 ± 0.19 |
| + Two-stage SFT & RL | 29.3 ± 0.70 | 47.1 ± 0.13 | 66.6 ± 0.53 | 53.0 ± 0.61 | 60.9 ± 0.75 | 51.4 ± 0.33 |
| + Interleaved SFT & RL | 29.2 ± 0.47 | 48.7 ± 0.54 | 71.8 ± 0.36 | 54.1 ± 0.72 | 64.3 ± 0.48 | 53.6 ± 0.10 |
| + Progressive SFT & RL | 29.8 ± 0.53 | 51.0 ± 0.30 | 72.4 ± 0.51 | 55.5 ± 0.82 | 65.9 ± 0.52 | 54.9 ± 0.22 |
| + Data Mixing | 29.2 ± 0.64 | 51.2 ± 0.26 | 72.0 ± 0.54 | 55.1 ± 0.97 | 62.7 ± 0.35 | 54.0 ± 0.40 |
| + Model Merging | 29.6 ± 0.17 | 50.4 ± 0.22 | 71.8 ± 0.67 | 53.7 ± 0.72 | 66.2 ± 0.46 | 54.3 ± 0.20 |

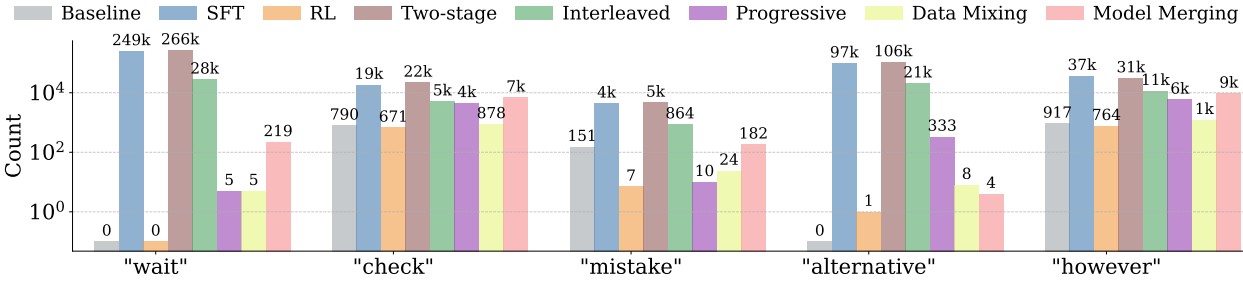

Figure 10: Frequency of reasoning words across fine-tuning methods, evaluated on the MathVision.

with our objective of leveraging the complementary strengths of SFT and RL. As shown in Tab. 6, the models accuracy consistently falls between that of SFT and RL across all benchmarks. Furthermore, Fig. 10 reveals a similar trend in the frequency of reasoning-related words. These results indicate that interleaved training achieves a balance between SFT and RL, rather than a complete synergy of their respective advantages.

**Progressive SFT & RL.** In previous methods involving SFT, the reasoning traces during training are entirely sourced from external models. However, during inference, the model generates each token based on self-generated prefix tokens, resulting a mismatch between training and inference. This inconsistency can be more pronounced in reasoning models that produce long responses. To alleviate this, we explore an alternative approach called progressive SFT and RL. Early in training, for difficult questions with a pass

rate of 0, the model relies on full external traces for SFT. As training goes on, the use of external traces is gradually reduced, starting with only a prefix and eventually removing them entirely. For loss computation, prefix tokens are assigned an SFT loss with a weight (empirically set to 0.2), while self-generated tokens are assigned an RL loss. As illustrated in Fig. 13, the training reward and validation accuracy curves of progressive training closely align with those of pure RL, while producing longer responses and exhibiting greater step-to-step fluctuations. As expected in Fig. 10, progressive training generates a higher frequency of reasoning-related words compared to pure RL. Benchmark results in Tab. 6 indicate that progressive training outperforms two-stage and interleaved training, with the average accuracy improving from 51.4 and 53.6 to 54.9, approaching RLs 55.1. Notably, progressive training achieves an accuracy of 29.8 on MathVision, surpassing RLs 29.0 and almost the same as SFTs 29.7. However, this improvement comes at a cost: progressive training sacrifices performance on MathVerse and MMStar, ultimately falling short of achieving the desired synergy.

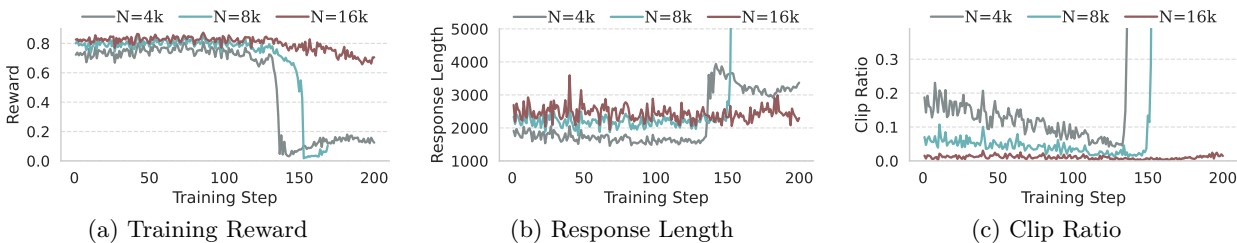

(a) Training Reward       (b) Response Length       (c) Clip Ratio

Figure 11: Training dynamics of RL following SFT. The model demonstrates a higher clip ratio when the maximum new token length is set to N=4k and 8k compared to N=16k, and collapses after approximately 100 steps.

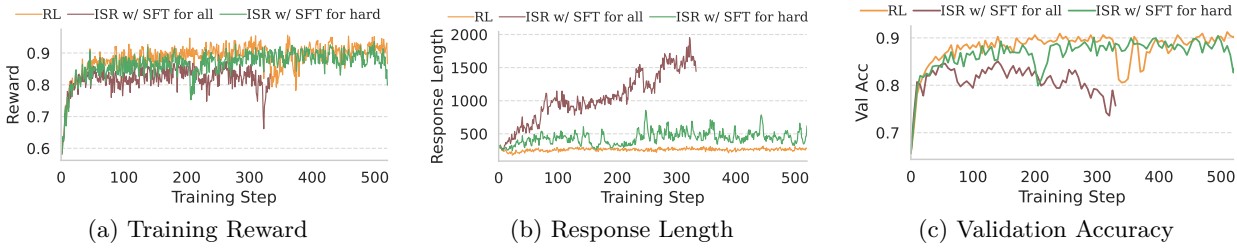

(a) Training Reward       (b) Response Length       (c) Validation Accuracy

Figure 12: Training dynamics of Interleaved SFT and RL (ISR) applied to all samples or with SFT loss only for hard questions. Pure RL is compared as the baseline.

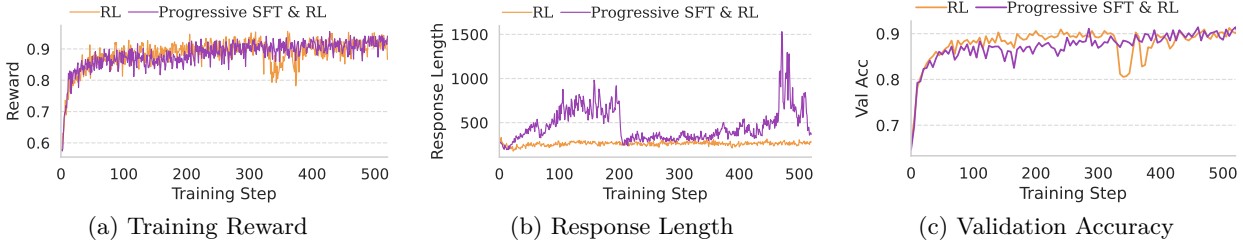

(a) Training Reward       (b) Response Length       (c) Validation Accuracy

Figure 13: Training dynamics of Progressive SFT and RL, compared to pure RL training.

## 3.2 Data Mixing

Data mixing is a simpler alternative to the above training alternation. It blends data distilled from SFT and RL models, followed by an additional round of SFT. In this method, we generate 8 responses per question using the RL model on the 34k Eureka-Distill dataset. We collect only the correct ones, getting 230k samples. For questions with a pass rate of 0, we add long-CoT responses, creating a final dataset of 243k samples. The

model is then trained on this dataset for 2 epochs, with all other settings consistent with those described in Sec. 2.1. As shown in Fig. 10, data mixing encourages more frequent use of reasoning tokens. However, it results in a 3.8% accuracy drop on MMStar and unexpectedly produces responses that are 10 times longer, as shown in Tab. 7. While data mixing demonstrates potential in fostering adaptive reasoning, its overall accuracy remains lower than that of pure RL.

Table 7: Comparison of model response length and accuracy between RL and data mixing approach.

|  | Model | MathVision | MathVerse | MathVista | MMMUval | MMStar | Avg. |
|---|---|---|---|---|---|---|---|
| Length | RL | 514 | 338 | 228 | 416 | 187 | 337 |
|  | Data Mixing | 637 | 381 | 324 | 424 | 1999 | 753 |
| Accuracy | RL | 29.0 | 52.1 | 72.6 | 55.1 | 66.5 | 55.1 |
|  | Data Mixing | 29.2 | 51.2 | 72.0 | 55.1 | 62.7 | 54.0 |

### 3.3 Model Merging

Model merging is a training-free approach that combines the weights of two or more models to harness their individual strengths. This technique typically involves interpolating the parameters of pre-trained models. We conduct experiments using MergeKit (Goddard et al., 2024), a versatile open-source toolkit for model merging. We adopt three model merging methods:

- **Linear Merging** was introduced in the "model soups" approach by Wortsman et al. (2022), which calculates a straightforward weighted average of the models' parameters.

- **TIES Merging** sparsifies the parameter changes (task vectors) and resolves sign disagreements before averaging (Yadav et al., 2023).

- **SLERP Merging** performs Spherical Linear Interpolation in the weight space between two models, enabling smooth transitions and preserving geometric consistency in the parameter space (Shoemake, 1985).

For SFT and RL models, we adjust the merging ratio of the SFT model, testing values of 0, 0.25, 0.5, 0.75, and 1. A ratio of 0 corresponds to the pure RL model, while a ratio of 1 represents the pure SFT model. Results are visualized in Fig. 14. Using the TIES method, SFT and RL models exhibit compatibility issues. In contrast, the Linear and SLERP methods show performance interpolation between SFT and RL, with a roughly monotonic increase on MathVision and a decrease on the other four benchmarks. The checkpoint with the highest average accuracy is achieved using the Linear method with an SFT ratio of 0.25. As shown in Tab. 6, this configuration preserves the high accuracy of the SFT model on MathVision and the high accuracy of the RL model on MMStar, while performing less competitively on the remaining datasets. These findings underscore the persistent challenges in achieving effective synergy between SFT and RL models.

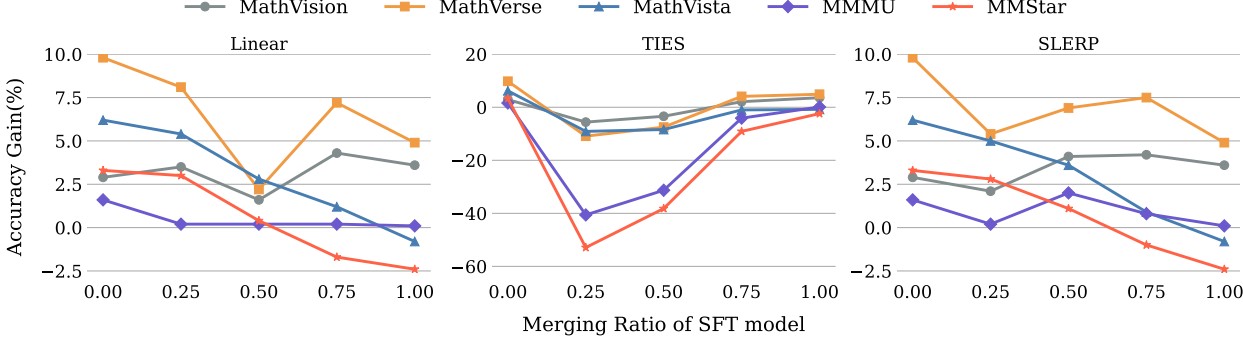

Figure 14: Accuracy gains of model merging methods (Linear, TIES, SLERP) by merging ratio.

### 3.4 Generalizing to Models of Other Sizes and Families

To validate the generalizability of our prior conclusions to other model sizes and families, we employed Qwen2.5-VL-3B and Gemma3-4B, trained via diverse post-training methods, and evaluated them on Math-Vision and MMStar (Tabs. 8, 9). Consistent with earlier findings, RL achieved the best overall performance, while SFT yielded limited, unstable gains. Among SFT-RL fusion strategies, Progressive training and model merging performed nearly as well as pure RL. Notably, interleaved training was unsatisfactory, revealing conflicts between fast and slow thinking training and associated stability risks. Further analysis of problem difficulty Fig. 15 shows the two models align with Qwen2.5-VL-7B: SFTs accuracy gains rise from easy to hard questions, outperforming RL on the hardest tasks but degrading by around 30% on simple ones, whereas RL delivers more balanced, stable improvements across all difficulty levels.

Table 8: Acuracy comparisons with Qwen2.5VL-3B.

| Model | MathVision | MMStar | Avg. |
|---|---|---|---|
| Qwen2.5-VL-3B | 21.3 ± 0.68 | 53.4 ± 0.70 | 37.4 ± 0.64 |
| + SFT | 21.8 ± 0.68 | 55.3 ± 0.98 | 38.5 ± 0.71 |
| + RL | 26.5 ± 0.67 | 59.4 ± 0.54 | 42.9 ± 0.56 |
| + Two-stage | 21.6 ± 0.90 | 55.1 ± 0.54 | 38.3 ± 0.44 |
| + Interleaved | 19.4 ± 0.59 | 49.7 ± 1.49 | 34.5 ± 0.96 |
| + Progressive | 24.0 ± 0.35 | 58.5 ± 0.27 | 41.2 ± 0.20 |
| + Data Mixing | 25.2 ± 0.40 | 57.9 ± 0.20 | 41.6 ± 0.21 |
| + Model Merging | 24.3 ± 0.33 | 58.1 ± 1.25 | 41.2 ± 0.74 |

Table 9: Acuracy comparisons with Gemma3-4B.

| Model | MathVision | MMStar | Avg. |
|---|---|---|---|
| Gemma3-4B | 22.9 ± 0.40 | 47.6 ± 0.27 | 35.3 ± 0.33 |
| + SFT | 20.7 ± 0.45 | 47.9 ± 1.81 | 34.3 ± 0.79 |
| + RL | 27.4 ± 0.55 | 50.8 ± 0.90 | 39.1 ± 0.61 |
| + Two-stage | 22.3 ± 0.47 | 51.0 ± 0.33 | 36.6 ± 0.31 |
| + Interleaved | 21.2 ± 0.27 | 45.8 ± 1.47 | 33.5 ± 0.67 |
| + Progressive | 26.4 ± 0.61 | 51.7 ± 0.45 | 39.0 ± 0.23 |
| + Data Mixing | 22.2 ± 0.47 | 49.8 ± 0.66 | 36.0 ± 0.46 |
| + Model Merging | 25.6 ± 0.52 | 52.2 ± 1.32 | 38.9 ± 0.71 |

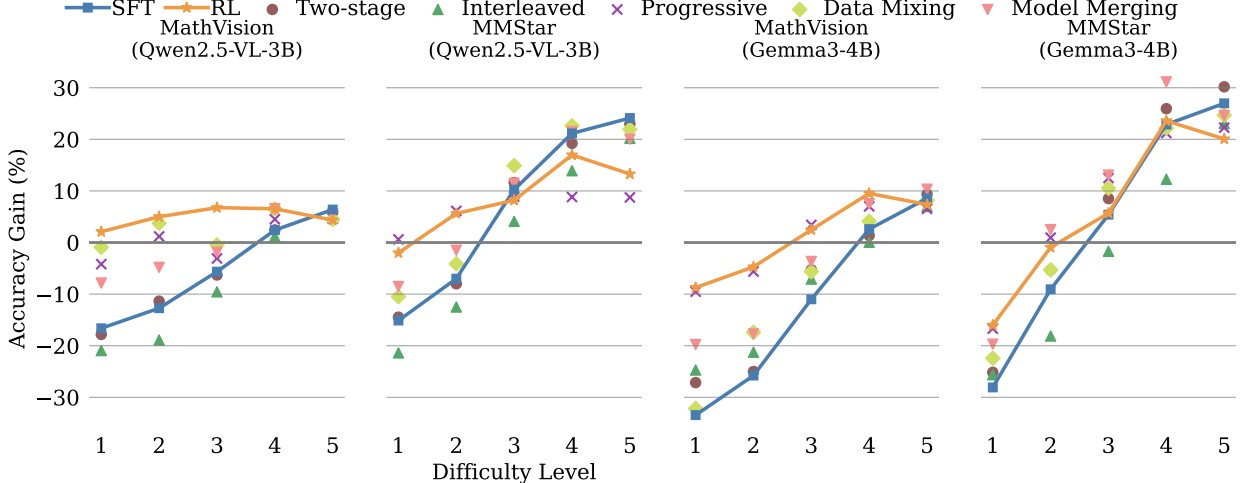

Figure 15: Accuracy gains for Qwen2.5-VL-3b and Gemma3-4b across five difficulty levels.

## 4 Related Work

The remarkable success of language reasoning models, such as OpenAI o1 (Jaech et al., 2024) and DeepSeek-R1 (Guo et al., 2025), has sparked significant interest in improving the reasoning capabilities of vision-language models (VLMs). These advancements are predominantly driven by novel post-training techniques, with supervised fine-tuning (SFT) and reinforcement learning (RL) as the two common and key approaches.

Early efforts focused on leveraging SFT to enhance long-form chain-of-thought (CoT) reasoning in VLMs. For example, LLaVA-CoT (Xu et al., 2024) introduced a structured reasoning dataset that guides models through sequential stages of summarization, visual interpretation, logical reasoning, and conclusion generation. Mulberry (Yao et al., 2024) utilized the collective knowledge of multiple models to collaboratively hypothesize, search, and identify effective reasoning trajectories leading to correct answers. Addi-

tionally, large-scale rewriting of original responses into CoT-style rationales using off-the-shelf models has been shown to significantly improve VLM reasoning performance (Guo et al., 2024). On the other hand, RL-based approaches have also demonstrated remarkable success in enhancing reasoning capabilities. MM-EUREKA (Meng et al., 2025) introduced the MMK12 dataset and a two-stage training strategy to stabilize RL training. VisualThinker-R1-Zero (Zhou et al., 2025) replicated R1-Zeros "Aha Moment" for multimodal reasoning on non-SFT VLMs. VLAA-Thinking (Chen et al., 2025a) challenged the effectiveness of SFT for cross-modal reasoning transfer and showed that training GRPO with a mixed-reward objective yields superior performance. Vl-rethinker (Wang et al., 2025a) introduced Selective Sample Replay (SSR) and Forced Rethinking techniques to address the vanishing advantages problem and encourage slow-thinking during reasoning.

Many works have explored combining SFT and RL, particularly within a two-stage training paradigm. These approaches differ primarily in their fine-tuning dataset design and RL strategies. LLaVA-Reasoner (Zhang et al., 2024b) distilled rationales from GPT-4o and applied Direct Preference Optimization (Rafailov et al., 2023) to improve CoT reasoning and generalization. Vision-R1 (Huang et al., 2025) performed SFT using a synthetic multimodal CoT dataset generated via modality bridging and trained GRPO with their Progressive Thinking Suppression Training (PTST) strategy. R1-VL (Zhang et al., 2025) proposed StepGRPO, an RL framework that rewards step-wise accuracy and logical validity rather than merely imitating correct reasoning paths. Reason-rft (Tan et al., 2025) incorporated three distinct types of accuracy rewards during RL training. Additionally, several studies have emphasized the importance of SFT data construction, systematically evaluating dataset generation methods to support effective RL from a cold start (Yang et al., 2025b; Wei et al., 2025; Chen et al., 2025d; Wen et al., 2025; Shen et al., 2025). In this work, we present a systematic analysis of the performance and behavior of VLMs trained exclusively with either SFT or RL. Furthermore, we explore various approaches to combine the strengths of SFT and RL from multiple perspectives, including training strategies, data mixing, and model merging.

## 5 Summary and Implications for Future Study

This study provides a systematic investigation into the roles of long-CoT SFT and RL in enhancing the reasoning capabilities of VLMs. Long-CoT SFT demonstrates strong performance on complex problems by introducing structured, step-by-step reasoning but suffers from verbosity and reduced accuracy on simpler tasks. RL on the other hand, promotes concise responses and robust generalization, delivering consistent gains across varying difficulty levels, though its gains on the hardest questions are lower than those from SFT. Attempts to combine SFT and RL, through two-stage, interleaved, and progressive training, as well as data mixing and model merging, all reveal a persistent "synergy dilemma", where trade-offs in accuracy, reasoning style, and response length dominate, preventing true complementarity.

We summarize below the key factors explaining why SFT and RL failed to synergize in our attempts, supported by experimental evidence. First, unlike sufficiently large models (e.g., >100B-parameter models such as DeepSeek-R1 and Kimi-k1.5) that use self-distilled long-CoT data for high compatibility and low overfitting, the relatively smaller models employed in this paper rely on distillation from external models. This reduces data-model compatibility, causes SFT-induced overfitting that RL cannot mitigate, and results in no improvement when combining SFT and RL compared to standalone SFT. Second, integrating SFTs structured "slow thinking" and RL's concise "fast thinking" requires adaptive switching between distinct reasoning styles, which is more complex than focusing on one mode. This results in trade-offs: SFT improves performance on hard questions but degrades it on easy ones, RL performs steadily across all difficulty levels but lags behind on the hardest questions, and hybrid strategies only balance these strengths without surpassing RL.

To address these challenges, future research should prioritize: 1) constructing model-compatible or self-distilled long-CoT datasets to mitigate data-model incompatibility or catastrophic forgetting after fine-tuning, using techniques such as prompt engineering and in-context learning; and 2) developing adaptive frameworks capable of accurately identifying problem difficulty, selecting optimal reasoning modes, and avoiding interference between different reasoning patterns. By overcoming these obstacles, VLMs can evolve into truly versatile models capable of reasoning efficiently and effectively across diverse multimodal tasks.

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

# A  Appendix

## A.1  Analysis with More Fine-Grained Separation of Difficulty Levels

To increse transparency and granularity of the 5-level difficulty categorization and validate that core trends are not artifacts of discrete binning, we supplement the main analysis with 17-point quasi-continuous results, where one data point stands for each possible pass rate of the baseline model (Qwen2.5-VL-7B) across 16 independent runs, ranging from 0/16 (extremely hard, baseline never solves) to 16/16 (trivially easy, baseline always solves). These results are visualized in Fig. 16, 17 , where each point represents the average accuracy gain of a post-training method (long-CoT SFT, RL, two-stage training, etc.) relative to the baseline, calculated for all benchmark questions falling into that specific pass rate bin.

Consistent with the 5-level analysis in Fig.1, the 17-point curves confirm three key trends: (1) For long-CoT SFT, accuracy gains transition from negative (for level 1 to level 7, corresponding to L1 easy questions) to positive as the difficulty level increases, with the largest gains concentrated at level 15 to level 17 (aligning with L4L5 hard/extremely hard questions); (2) RL maintains more steady positive gains across level 3 to

level 17, confirming its strength in generalizing to both easy and hard tasks; (3) Hybrid strategies (e.g., interleaved training, model merging) continue to exhibit trade-offs rather than synergy: their curves hover between the SFT and RL curves for most pass rates, never combining SFTs high gains at low pass rates with RLs consistent gains at high pass rates.

Notably, the 17-point results exhibit slightly more fluctuations than the 5-level analysis, which is expected given the smaller sample size per individual pass rate bin (e.g., fewer questions have an exact pass rate of 3/16 or 14/16, compared to the aggregated L3 or L1 bins). Despite this noise, the consistency in directional trends across both discrete and continuous representations confirms that the papers findings and conclusions, including SFTs difficulty-dependent trade-off, RLs broad generalization, and the SFT-RL synergy dilemmaare robust to the choice of difficulty granularity.

## A.2 Remarks on the Differences to Leaderboard Results

The performance of Qwen2.5-VL-7B, as reported in our paper, is slightly lower than the performance on the Open VLM Leaderboard[1] across the MMStar, MathVista, and MMMU val benchmarks. We clarify that this slight discrepancy stems from minor variations in evaluation settings, which do not alter the core findings and conclusions of this paper. We have taken steps to align our experimental setup with the leaderboard as much as possible: we adopted GPT-4o-mini as the judge for MathVista and MMMU val evaluations, and GPT-4-0125 for MMStar, consistent with the leaderboards configuration. Additionally, we observed that the leaderboard results are derived from a single run (likely the optimal result selected from multiple attempts). Likewise, we reproduced the baseline performance of Qwen2.5-VL-7B by taking the maximum accuracy across 4 independent runs. This maximum value is highly consistent with the leaderboard results (see Tab. 10). For robustness against random errors, the main results reported in the paper are the average accuracy across 4 runs, which is slightly lower than the maximum accuracy as expected.

To reduce the cost of GPT API calls, we used the locally deployed Qwen2.5-VL-32B as an alternative judge. Although the absolute accuracy values obtained with the Qwen judge are slightly lower than those with GPT judges, the trends in performance gains from SFT and RL remain consistent across both judge settings. Specifically, when using GPT as the judge, SFT leads to an average accuracy drop of 1.2% and RL leads to an average gain of 3.0%; when using Qwen2.5-VL-32B as the judge, SFT results in a 1.0% average drop and RL in a 3.7% average gain (see Tab. 11). This consistency confirms that the impact of SFT and RL on model performance is not dependent on the choice of judge, further validating our conclusions.

Table 10: Comparison of Qwen2.5-VL-7B Performance Across Different Evaluation Settings.

| Qwen2.5-VL-7B | Judge | MMStar | MathVista | MMMU val | Avg. |
|---|---|---|---|---|---|
| Reported on Open VLM Leaderboard | GPT | 64.1 | 68.1 | 58.0 | 63.4 |
| Reproduced (Max. of 4 runs) | GPT | 64.1 | 68.3 | 58.6 | 63.7 |
| Reproduced (Avg. of 4 runs) | GPT | 63.5 | 67.5 | 57.8 | 62.9 |

Table 11: Performance Gains of SFT and RL Under Different Judges.

| Judge | Model | MMStar | MathVista | MMMU val | Avg. |
|---|---|---|---|---|---|
| GPT | Qwen2.5-VL-7B | 63.5 | 67.5 | 57.8 | 62.9 |
| | + SFT | 60.7 | 66.2 | 58.1 | 61.7 (-1.2%) |
| | + RL | 66.5 | 72.6 | 58.7 | 65.9 (+3.0%) |
| Qwen2.5-VL-32B | Qwen2.5-VL-7B | 63.2 | 66.4 | 53.5 | 61.0 |
| | + SFT | 60.8 | 65.6 | 53.6 | 60.0 (-1.0%) |
| | + RL | 66.5 | 72.6 | 55.1 | 64.7 (+3.7%) |

---

[1]Open VLM Leaderboard. `https://huggingface.co/spaces/opencompass/open_vlm_leaderboard`

### A.3 Attempts to Elicit Long-CoT with Prompt Engineering

We explored prompt engineering techniques to elicit long, structured Long-CoT reasoning traces from our target models. Specifically, we tested a range of strategies: zero-shot prompts (e.g., "Think through this step by step in detail"), few-shot examples with verbose reasoning demonstrations, and iterative prompting to encourage extended elaboration. However, these efforts yielded limited success: the models consistently produced concise, task-oriented responses rather than the detailed, multi-step reasoning we aimed for.

We attribute this to two constraints: 1) The tested models ($\leq$ 7B parameters) lack the scale ($\geq$ 100B) needed to generate complex, high-fidelity Long-CoT traces. 2) Being instruction-tuned, they are optimized for direct, coherent responses aligned with instruction-following goals, which conflicts with the unconstrained verbosity required for Long-CoT. This contrasts with base models like DeepSeek-R1, which retain flexibility to produce the "aha moments" and extended reasoning seen in self-distilled Long-CoT data.

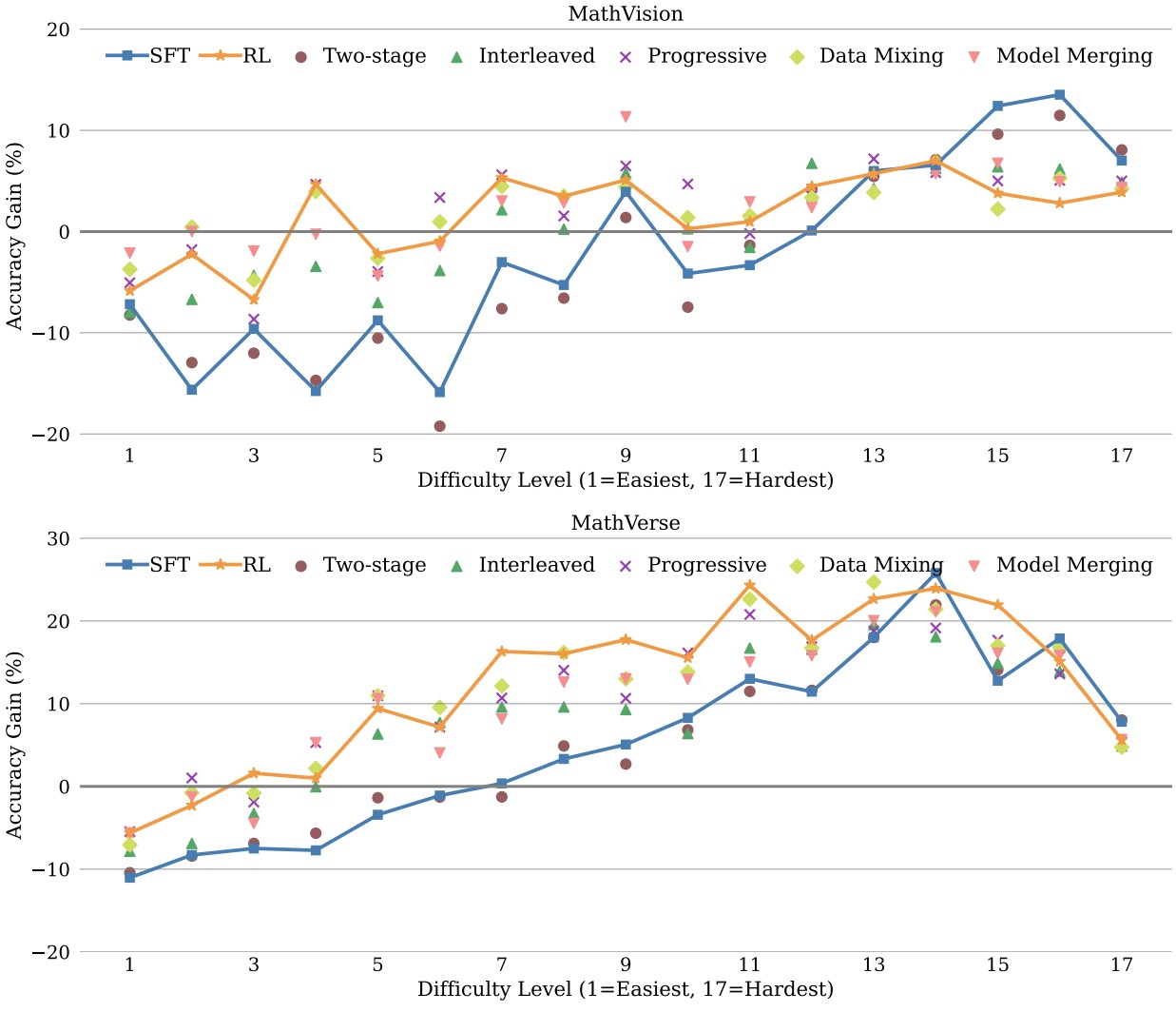

Figure 16: Accuracy gains from various post-training techniques across 17 difficulty levels.

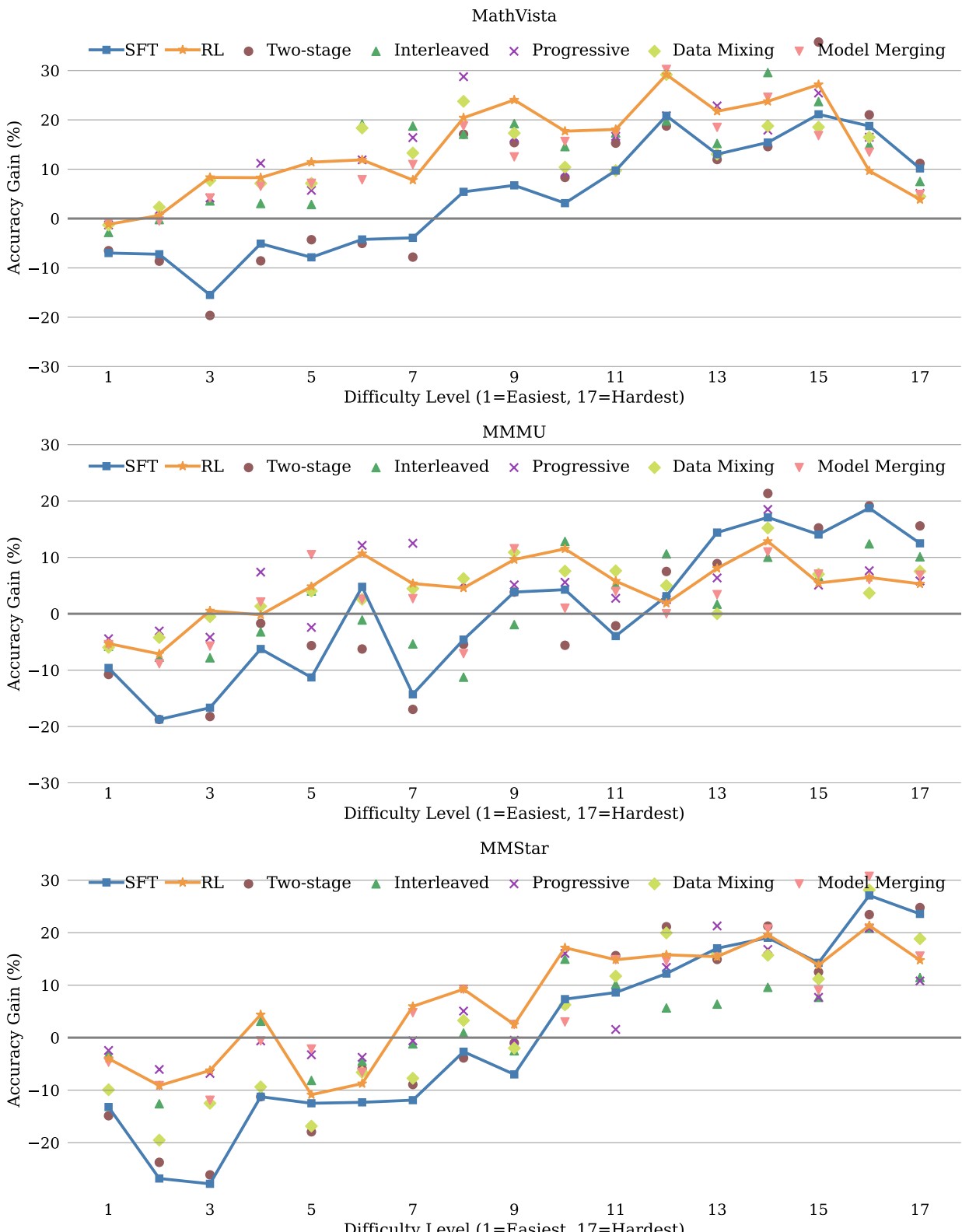

Figure 17: Accuracy gains from various post-training techniques across 17 difficulty levels (continued).

