# OpenReview forum: "The Synergy Dilemma of Long-CoT SFT and RL: Investigating Post-Training Techniques for Reasoning VLMs"
_TMLR — Accepted by TMLR_

### Review · Reviewer_Teh3 · 2025-08-03

**Summary Of Contributions:**

This paper compares finetuning of multimodal reasoning models through Reinforcement Learning (RL) and Supervised Finetuning (SFT). By fine-tuning Qwen2.5-VL-7B, the paper finds that SFT only achieves performance gains on the hardest problems, while RL achieves performance gains on all difficulty levels, but less pronounced ones than SFT on hard problems. Experiments on how to combine both training strategies yield negative results.

**Audience:**

Yes

**Audience Explanation:**

The research questions asked in this paper are very important, and providing answers could be very impactful. Finding ways to effectively combine RL and SFT, as well as understanding their respective strengths and weaknesses, has important implications for training stronger and more generalizable models.

**Claims And Evidence:**

No

**Claims Explanation:**

The paper promises a systematic investigation into the impact of SFT and RL on long-form reasoning in MLLMs. However, the experimental setup of the paper is limited in the following aspects:

**(L1)** All experiments involve only one model, `Qwen2.5-VL-7B`. However, to show that results generalize, it is necessary to test multiple models. For example, experiments could be conducted with 2 models of different sizes (e.g. 2B vs. 7B) using models from 2 different model families (e.g. Llava vs. Qwen-VL) .

**(L2)** The "systematicity" of the study is not properly defined. For this, I would expect a clear definition of which axes of variation are evaluated, and why they are the most relevant. It is unclear if the experiments on combining SFT and RL are a collection of potential solutions or follow any structure, e.g. one derived from a literature survey or accepted taxonomy of the field.

**(L3)** The performance gains achieved through finetuning (both SFT and RL) are consistently below the online leaderboard performance of Qwen2.5-VL-7B [a] (concerning MMStar, MathVista, and MMMU val). Since no confidence intervals or similar measures are reported, it is unclear how significant the reported results are.

**(L4)** The categorization of problems into difficulty levels is not well motivated, and the choice of passing rate intervals in Sec. 2.3 is not explained. For example, instead of using discrete difficulty levels, these results could be represented as a curve showing the accuracy on the problems that pass at least $k$ times, showing 16 points in the proposed setup. This would allow a more transparent assessment.

[a] https://huggingface.co/spaces/opencompass/open_vlm_leaderboard

**Requested Changes:**

To improve the validity and show the reported insights generalize, the paper should
 * Show that findings hold when varying model properties along meaningful axes of variations, such as model size and model family (as a proxy for training data and objective)
 * Clearly define which aspects of the study are "systematic" and what is the underlying system from which experimental variations are derived
 * Strengthen the validity by reporting confidence intervals, error barss, or p-values; or alternatively provide justification why the reported performance differences are statistically significant.
 * Explain the differences to leaderboard performance
 * Better explain the categorization of problems into difficulty levels, or present results in a way that is independent of such classification

I believe that these changes are necessary to prove the validity of the presented conclusions, and including these recommendations will significantly strengthen the paper.

---

> ### Author Response · Authors · 2025-11-06
> **Response (1/2)**
>
> **1: Models of different sizes and families**
>
> Thank you for your helpful advice. We have following your advice to validate the generalizability of our prior conclusions to other model sizes and families. We employed Qwen2.5-VL-3B and Gemma3-4B, trained via diverse post-training methods, and evaluated them on MathVision and MMStar (Tabs.8, 9). **Consistent with earlier findings**, RL achieved the best overall performance, while SFT yielded limited, unstable gains. Among SFT-RL fusion strategies, Progressive training and model merging performed nearly as well as pure RL. Notably, interleaved training was unsatisfactory, revealing conflicts between fast and slow thinking training and associated stability risks. Further analysis of problem difficulty Fig.15 shows the two models align with Qwen2.5-VL-7B: SFT’s accuracy gains rise from easy to hard questions, outperforming RL on the hardest tasks but degrading by around 30\% on simple ones, whereas RL delivers more balanced, stable improvements across all difficulty levels.
>
> Accordingly, we have incorporated these additional results and discusstions in Sec. 3.4.
>
>
> **2: The "systematicity" of the study**
>
> The study’s "systematicity" is reflected in three axes:
>
> - **Diverse Post-Training Paradigms**: Evaluates standalone methods (SFT leveraging external knowledge, RL leveraging internal knowledge) and three fusion strategies including pre-training data mixing, training-time fusion, and post-training model merging. These approaches cover the dominant technical paths identified in our related work (Sec. 4), avoiding arbitrary trial designs.
> - **Comprehensive Evaluation**: Assesses performance across 5 benchmarks (e.g., MathVision, MathVerse), efficiency via response length, and reasoning style (e.g., frequency of key reasoning words).
> - **Varied Models**: Following revisions, we use 3 model sizes (3B, 4B, 7B) and 2 model families (Qwen2.5-VL, Gemma3). This design ensures our findings generalize across different model architectures and scales.
>
> Accordingly, we have updated the introduction in Sec. 1.
>
>
>
> **3. Confidence intervals**
>
> Thanks for your suggestion! We have updated Table 6 by further reporting the 95% confidence intervals, which demonstrate the significant differences among various training paradigms and the validity of our findings.
>
>
> **4: Explain the differences to leaderboard performance**
>
> Thanks for your thorough review. The slight difference between our scores and those on the leaderboard mainly stems from **minor variations in evaluation settings. But this doesn’t change the findings and conclusions in the paper.**
>
> First, we managed to align with the settings of the leaderboard: we used GPT-4o-mini as the judge for evaluating MathVista and MMMU val, and GPT-4-0125 as the judge for evaluating MMStar. Additionally, we noticed that the results on the leaderboard are from a single run, which is likely the best result selected from several runs. Therefore, when we reproduced the baseline results of Qwen2.5VL-7B, we chose the maximum accuracy from 4 runs (as shown in the table below), and this result is consistent with the leaderboard.
>
> To minimize the impact of random errors, the values we mainly report in the paper are the average accuracy of 4 runs. As shown in the table below, this value is slightly lower than the maximum accuracy, as expected.
>
>
> | Qwen2.5VL-7B | Judge | MMStar |  MathVista | MMMU val | Avg  |
> |:-------------|:--------------:|:--------------:|:--------------:|:--------------:|:--------------:|
> | Reported on OpenVLM leaderboard       | GPT | 64.1 | 68.1 | 58.0 | 63.4
> | Reproduced, Max. of 4 runs   | GPT | 64.1 | 68.3 | 58.6 | 63.7
> | Reproduced, Avg. of 4 runs   | GPT | 63.5 | 67.5 | 57.8 | 62.9
>
> Furthermore, to reduce the cost of using the GPT API, we opted for the locally deployed model Qwen2.5-VL-32B as the judge. Although the evaluation results here are slightly lower than those obtained with GPT as the judge, the trends in measuring the impact of SFT and RL on the baseline model are consistent whether GPT or Qwen is used as the judge. As shown in the table below: when using GPT as the judge, the average accuracy gains of SFT and RL are -1.2% and +3.0% respectively; when using Qwen as the judge, the average accuracy gains of SFT and RL are -1.0% and +3.7% respectively.
>
>
> | Judge  | Model | MMStar |  MathVista | MMMU val | Avg  |
> |:-------------|:--------------:|:--------------:|:--------------:|:--------------:|:--------------:|
> | GPT | Qwen2.5VL-7B   | 63.5 | 67.5 | 57.8 | 62.9
> |     | + SFT          | 60.7 | 66.2 | 58.1 | 61.7 (-1.2)
> |     | + RL           | 66.5 | 72.6 | 58.7 | 65.9 (+3.0)
> | Qwen | Qwen2.5VL-7B  | 63.2 | 66.4 | 53.5 | 61.0
> |     | + SFT          | 60.8 | 65.6 | 53.6 | 60.0 (-1.0)
> |     | + RL           | 66.5 | 72.6 | 55.1 | 64.7 (+3.7)
>
> Accordingly, we have incorporated the above clarifications in the appendix (see Sec. A.2)

---

> ### Author Response · Authors · 2025-11-06
> **Response (2/2)**
>
> **5: Motivation for difficulty levels, and the choice of passing rate intervals**
>
> Thank you for raising this question. The primary goal of categorizing problems by difficulty is to **resolve the "average performance ambiguity"** in evaluating long-CoT SFT and RL for VLMs. As noted in the paper (Sec. 1, Sec. 2.3), multimodal reasoning benchmarks (e.g., MathVista, MMMU) contain a mix of simple perception tasks (e.g., recognizing objects in images) and complex cognitive reasoning tasks (e.g., solving visual math problems requiring multi-step logic). Without difficulty stratification, the overall average accuracy would mask critical tradeoffs and the unique strengths of each method, e.g., SFT improves hard problems but harms easy ones, while RL is consistent across all difficulties (see Fig. 7).
>
> Regarding the pass rate intervals, we use 5 levels to **balance granularity and statistical reliability**. This ensures an adequate sample size per level, which is critical for reliable accuracy estimates. For greater transparency, we **have also included 17-point quasi-continuous results** (pass rates from 0/16 as the hardest level 17 to 16/16 as the easiest level 1) in the appendix (see Figs. 16 and 17). These figures display trends consistent with those in Fig. 1 (using 5-level categorization), albeit with more fluctuations.
>
> Accordingly, we have explained how problems are categorized into difficulty levels in the appendix (see Sec. A.1).

---

### Review · Reviewer_XFAT · 2025-08-30

**Summary Of Contributions:**

Summary:

This paper examines the roles of two key steps (long-cot SFT and RL) in the VLM post-training stage. Authors explored the distinct results and the combined results from three different perspectives and found interesting conclusions - the synergy dilemma of long-cot SFT and RL, which shows combining two methods does not lead to performance gains. Additionally, authors also investigate through detailed analysis, such as token KL divergence.

Strengths:

1. The paper structure and writing are clear and easy to understand. The figures are clear and sufficient.
2. The result comparison is extensive and fair.

Weaknesses:

1. The datasets and models selected are insufficient to demonstrate the conclusion. More datasets and models at different scales are needed.
2. Paper lacks more analysis, such as the key reason why VLMs behave differently from LLMs.

**Audience:**

Yes

**Audience Explanation:**

The research question of the paper is still interesting and provides a data point that I believe can foster future works, despite its weaknesses.

**Claims And Evidence:**

No

**Claims Explanation:**

As mentioned, to demonstrate the existence of the synergy dilemma in VLMs (a relatively serious claim), more datasets and models at different scales are needed.

**Requested Changes:**

If compute resources allow, please address the aforementioned weaknesses.

A few more questions that I am interested in and might be added to the manuscript:

1. The core difference that makes VLMs behave differently from LLMs.
2. If using rejection sampling instead of the Eureka-Distill dataset, will the model still exhibit the verbose reasoning process? Typically, RL would elicit a surge in response length and 'aha' words while the opposite is observed in this paper.
3. According to the results, for the L1 questions, almost all methods lead to a performance drop. What is the reason behind?
4. For the model collapse during RL training, what could be the reasons behind?

---

> ### Author Response · Authors · 2025-11-06
> **Response (1/2)**
>
> **1: Models and datasets**
>
> Thank you for your helpful advice. We have following your advice to validate the generalizability of our prior conclusions to other model sizes and families. We employed Qwen2.5-VL-3B and Gemma3-4B, trained via diverse post-training methods, and evaluated them on MathVision and MMStar (Tabs.8, 9). **Consistent with earlier findings**, RL achieved the best overall performance, while SFT yielded limited, unstable gains. Among SFT-RL fusion strategies, Progressive training and model merging performed nearly as well as pure RL. Notably, interleaved training was unsatisfactory, revealing conflicts between fast and slow thinking training and associated stability risks. Further analysis of problem difficulty Fig.15 shows the two models align with Qwen2.5-VL-7B: SFT’s accuracy gains rise from easy to hard questions, outperforming RL on the hardest tasks but degrading by around 30\% on simple ones, whereas RL delivers more balanced, stable improvements across all difficulty levels.
>
> Accordingly, we have incorporated these additional results and discusstions in Sec. 3.4.
>
> Regarding datasets, we tested SFT datasets distilled from Gemini2 and DeepSeekR1 (see Tab. 1). We also explored additional datasets including LIMO[1], R1-Onevision[2], and Virgo[3], but these yielded low data-model compatibility; we thus report the best SFT results in this paper. The Eureka-Distilled dataset for the main experiments are derived from the high-quality, diverse Eureka dataset[4]. For benchmark datasets, we adopted those widely used in the field, including those employed to evaluate baseline models in their original studies.
>
> [1] LIMO: Less is More for Reasoning \
> [2] R1-Onevision: Advancing Generalized Multimodal Reasoning through Cross-Modal Formalization \
> [3] Virgo: A Preliminary Exploration on Reproducing o1-like MLLM \
> [4] MM-Eureka: Exploring the Frontiers of Multimodal Reasoning with Rule-based Reinforcement Learning \
>
> **2: Why VLMs behave differently from LLMs**
>
> Thanks for raising the question. As we have discussed the **benchmark characteristics** in the introduction, textual reasoning benchmarks (e.g., AIME25, GPQA) focus on complex math or logical reasoning, while multimodal reasoning benchmarks (MathVista, MMMU, MMStar) contain a large proportion of simple perception/visual understanding questions. This creates a tension: SFT’s verbosity degrades performance on simple questions, while RL’s brevity limits depth on hard multimodal tasks, undermining synergy.
>
> Below we provided two more reasons explaining why SFT and RL failed to synergize in our attempts, supported by experimental evidence. First, unlike sufficiently large models (e.g., >100B-parameter models such as DeepSeek-R1 and Kimi-k1.5) that use self-distilled long-CoT data for high compatibility and low overfitting, the relatively smaller models employed in this paper rely on distillation from external models. This reduces data-model compatibility, causes SFT-induced overfitting that RL cannot mitigate, and results in no improvement when combining SFT and RL compared to standalone SFT. Second, integrating SFT’s structured ''slow thinking'' and RL's concise ''fast thinking'' requires adaptive switching between distinct reasoning styles, which is more complex than focusing on one mode. This results in trade-offs: SFT improves performance on hard questions but degrades it on easy ones, RL performs steadily across all difficulty levels but lags behind on the hardest questions, and hybrid strategies only balance these strengths without surpassing RL.
>
> We have updated the conclusion (Sec. 5) accordingly.
>
> **3: Using rejection sampling**
>
> We appreciate your insights. We did attempt rejection sampling to elicit structured, lengthy reasoning traces, but this approach proved ineffective. We hypothesize two key reasons:
> * The tested models are not sufficiently large (e.g., <100B), limiting their ability to generate high-quality reasoning traces.
> * All tested models are instruction-tuned, having undergone instruction-following training. This makes it less flexible in response styles and harder to elicit the "aha moments" observed in base models like DeepSeek-R1-base.
>
> Accordingly, we have incorporated these discusstions into the appendix (Sec. A.3).
>
> **4: Performance drop**
>
> We clarify that all post-training techniques outperform the baseline model (see Tab.6). The perceived "performance drop" refers to most lagging behind pure RL, which stems from distinct strengths: SFT excels at hard questions but causes verbosity on easy tasks, while RL optimizes for generalization/brevity across all difficulty levels, and hybrid methods struggle to balance these two, leading to no clear net gain over pure RL.

---

> ### Author Response · Authors · 2025-11-06
> **Response (2/2)**
>
> **5: Model collapse during RL training**
> As encouter during our RL training, model collapse is mainly attributed to two factors: **lack of KL divergence regularization** and **infinite loops** in reasoning traces.
> * Without KL divergence, the model loses constraints on deviation from the reference model. Unconstrained exploration causes unstable training dynamics, preventing convergence and leading to collapse (see Sec. 2.2).
> * Reasoning traces getting stuck in infinite loops trigger soaring response length, as shown in Fig.11 (b). Redundant/repetitive outputs make rewards ineffective for guiding learning, resulting in model collapse.

---

### Review · Reviewer_Jkqh · 2025-10-23

**Summary Of Contributions:**

The paper empirically compares long chain-of-thought (CoT) supervised fine-tuning (SFT) and reinforcement learning (RL) for reasoning in large vision-language models. SFT improves performance on hard problems via structured, step-by-step reasoning but tends to induce verbosity that can hurt easier tasks. RL yields more concise, broadly improved answers but offers smaller gains on the hardest questions. Combining the two—via two-stage, interleaved, and progressive training, data mixing, or model merging—reflects a “synergy dilemma”: rather than additive benefits, the combination leads to a trade-off between the two approaches.

**Audience:**

Yes

**Audience Explanation:**

The paper addresses a topical question—how long-CoT SFT compares with RL for reasoning in vision-language models—and reports negative/neutral results on their combination (the “synergy dilemma”). Given the current interest in multimodal research, this work may be of interest to parts of the community, even as a negative result.

**Claims And Evidence:**

Yes

**Claims Explanation:**

The claims are supported by experiments comparing long-CoT SFT and RL on five VLM benchmarks (MathVision, MathVerse, MathVista, MMMU-val, and MMStar-val) using the Qwen2.5-VL-7B model. The combination of the two is examined via training alternation (two-stage, interleaved, progressive), data mixing, and model merging. The evidence is consistent with the stated “synergy dilemma”: attempts to combine SFT and RL do not yield additive gains. The analyses (e.g., accuracy gains versus average length trade-offs) are presented clearly.
However, the explanation for why VLMs exhibit a “synergy dilemma” when combining long-CoT SFT and RL, in contrast to the improvements sometimes observed in LLMs, remains insufficiently clear and not fully convincing.

**Requested Changes:**

- Include at least one additional base model to test whether the conclusion still holds beyond Qwen2.5-VL-7B.
- Report results across multiple random seeds and assess statistical significance (e.g., by including variance or error bars).
- Add a brief paragraph explaining why VLMs exhibit a “synergy dilemma” when combining long-CoT SFT and RL, whereas some LLM papers report improvements; summarize plausible factors and indicate which are supported by your experiments.

---

> ### Author Response · Authors · 2025-11-06
> **Response (1/2)**
>
> **1: Additional base models**
>
> Thank you for your helpful advice. We have following your advice to validate the generalizability of our prior conclusions to other model sizes and families. We employed Qwen2.5-VL-3B and Gemma3-4B, trained via diverse post-training methods, and evaluated them on MathVision and MMStar (Tabs.8, 9). **Consistent with earlier findings**, RL achieved the best overall performance, while SFT yielded limited, unstable gains. Among SFT-RL fusion strategies, Progressive training and model merging performed nearly as well as pure RL. Notably, interleaved training was unsatisfactory, revealing conflicts between fast and slow thinking training and associated stability risks. Further analysis of problem difficulty Fig.15 shows the two models align with Qwen2.5-VL-7B: SFT’s accuracy gains rise from easy to hard questions, outperforming RL on the hardest tasks but degrading by around 30\% on simple ones, whereas RL delivers more balanced, stable improvements across all difficulty levels.
>
> Accordingly, we have incorporated these additional results and discusstions in Sec. 3.4.
>
>
> **2: Statistical significance**
>
> Thanks for your suggestion! In our original manuscript, we already reported all results using the mean values from four independent runs. In response to your suggestion, we now further include the 95% confidence intervals in Table 6, which demonstrate the significant differences among various training paradigms and the validity of our findings.

---

> ### Author Response · Authors · 2025-11-06
> **Response (2/2)**
>
> **3: Why “synergy dilemma” of long-CoT SFT and RL for VLMs**
>
> Below we discuss plausible factors explaining the discrepancy, with explicit alignment to our experimental findings:
> * **Data Source and Model Compatibility**. Sufficiently large language models (e.g., >100B like DeepSeek-R1 and Kimi-k1.5) often use self-distilled long-CoT data. This data is usually generated by the model itself, ensuring high compatibility and reducing overfitting risks. In contrast, smaller-scale vision-language models (like our Qwen2.5-VL-7B) struggle to produce high-quality long-CoT traces through self-distillation. In our experiments, we tried to distill long-cot sft data from various types of external models (either LLM or VLM, such as LIMO[1], R1-OneVision[2], Virgo[3]), all leading to low data-model compatibility and we put the best SFT results in our paper. This can cause SFT to induce overfitting to external reasoning patterns especially for VLMs, undermining the model’s inherent capabilities and limiting synergy with RL.
> Experimental Support: Confirmed by our two-stage training results (Tab. 6). The two-stage SFT+RL approach failed to improve over standalone SFT (average accuracy 51.4% vs. 51.4%), indicating SFT-induced overfitting that RL could not mitigate. Additionally, our SFT experiments showed that low-quality external data (e.g., s1-Gemini2) degraded performance despite longer responses (Tab. 1, Fig. 2), highlighting the critical role of data-model compatibility.
>
> [1] LIMO: Less is More for Reasoning
>
> [2] R1-Onevision: Advancing Generalized Multimodal Reasoning through Cross-Modal Formalization
>
> [3] Virgo: A Preliminary Exploration on Reproducing o1-like MLLM
>
> * **Compatibility of "Fast" vs. "Slow" Thinking Modes**. Our work aims to integrate SFT’s "slow thinking" (structured, verbose chain-of-thought for hard questions) and RL’s "fast thinking" (concise, generalized responses for all difficulty levels). This requires the model to adaptively switch between two distinct reasoning styles, a more complex goal than LLMs that focus on either mode alone. Incompatibility between these modes leads to trade-offs (e.g., verbosity vs. brevity) instead of synergy.
> Experimental Support: Directly validated by our comparative analysis (Sec. 2.3, Fig. 6, Fig. 8). SFT improved hard questions (L4-L5) but hurt easy ones (L1) due to overthinking; RL delivered steady gains but underperformed SFT on L5. Hybrid strategies (e.g., interleaved training) only balanced these strengths (average accuracy 53.6%) without surpassing RL, confirming conflicting reasoning modes (Fig. 10: reasoning word frequency of hybrids fell between SFT and RL).
>
> * **Benchmark Characteristics**. Textual reasoning benchmarks (e.g., AIME25, GPQA) focus on complex math or logical reasoning, while multimodal reasoning benchmarks (MathVista, MMMU, MMStar) contain a large proportion of simple perception/visual understanding questions. This creates a tension: SFT’s verbosity degrades performance on simple questions, while RL’s brevity limits depth on hard multimodal tasks, undermining synergy.
> Experimental Support: Explicitly supported by our difficulty-level analysis (Fig. 1). SFT’s accuracy gain was negative for L1 (easy) questions across benchmarks but positive for L4-L5 (hard); RL gained consistently across all levels but lagged SFT on L5. Hybrid methods failed to resolve this tension (e.g., progressive training improved pure RL on MathVision L5 but sacrificed MathVerse performance, Table 6), directly linking benchmark composition to the synergy dilemma.
>
> We have summarized the above into a brief paragraph, and included in the conclusion (Sec. 5).

---

### Decision · Action_Editor_6rr7 · 2025-12-15

**Recommendation:** Accept as is

**Audience:**

Yes

**Audience Explanation:**

The paper addresses an important question about long-CoT Supervised fine tuning  and RL for reasoning in vision-language models. The additional base models given by the authors along with the new experiments broadened the scope of the study which would be of interest for significative audience of the ML community.

**Claims And Evidence:**

Yes

**Claims Explanation:**

This paper presents an extensive empirical analysis of the distinct roles of  Long Chain-of-Thought Supervised Fine-Tuning (CoT SFT)  and Reinforcement Learning (RL) for multimodal reasoning tasks. The initial contribution was conducted using a fine-tuned version of Qwen2.5-VL-7B.Reviewers found the paper’s contribution interesting for the ML community  and appreciated the structure of the paper.

However, concerns were raised regarding the limited diversity of experimental settings, the need for additional base models to discuss the generality of the conclusions beyond Qwen2.5-VL-7B. They also wanted the authors to highlight the statistical significance of the results, and  to comment more on the performance of the approaches (gains from fine-tuning below some  state-of-the-art levels for instance, settings with occasional performance drops).

The authors substantially strengthened the empirical work and the discussions and motivations of the paper during the rebuttal phase. They incorporated additional models and provided clearer analyses. Specifically, they introduced Qwen2.5-VL-3B and Gemma3-4B, with  new results summarized in Section 3.4, thereby significantly broadening the scope of the study. A detailed appendix further elaborates on the authors choices and provides additional discussions of algorithmic performance. Although the scope remains somewhat constrained to few vision-language models, the methodology is sound and the results are promising and offer several insights for future research.